# The Influence of Lateral Size and Oxidation of Graphene Oxide on Its Chemical Reduction and Electrical Conductivity of Reduced Graphene Oxide

**DOI:** 10.3390/molecules27227840

**Published:** 2022-11-14

**Authors:** Hak Jin Sim, Zheling Li, Ping Xiao, Hui Lu

**Affiliations:** 1School of Biological Sciences, Faculty of Biology, Medicine and Health, The University of Manchester, Manchester M13 9PT, UK; 2Department of Materials, Faculty of Science and Engineering, The University of Manchester, Manchester M13 9PL, UK; 3Department of Materials, National Graphene Institute, Faculty of Science and Engineering, The University of Manchester, Manchester M13 9PL, UK; 4Department of Materials, Henry Royce Institute, Faculty of Science and Engineering, The University of Manchester, Manchester M13 9PL, UK

**Keywords:** graphene oxide, reduced graphene oxide, GO reduction efficiencies, electrical conductivity, graphene-related materials and films

## Abstract

The chemical reduction efficiencies of graphene oxide (GO) are critically important in achieving graphene-like properties in reduced graphene oxide (rGO). In this study, we assessed GO lateral size and its degree of oxidation effect on its chemical reduction efficiency in both suspension and film and the electrical conductivity of the corresponding rGO films. We show that while GO-reduction efficiency increases with the GO size of lower oxidation in suspension, the trend is opposite for film. FESEM, XRD, and Raman analyses reveal that the GO reduction efficiency in film is affected not only by GO size and degree of oxidation but also by its interlayer spacing (restacking) and the efficiency is tunable based on the use of mixed GO. Moreover, we show that the electrical conductivity of rGO films depends linearly on the C/O and Raman I_D_/I_G_ ratio of rGO and not the lateral size of GO. In this study, an optimal chemical reduction was achieved using premixed large and small GO (L/SGO) at a ratio of 3:1 (*w*/*w*). Consequently, the highest electrical conductivity of 85,283 S/m was achieved out of all rGO films reported so far. We hope that our findings may help to pave the way for a simple and scalable method to fabricate tunable, electrically conductive rGO films for electronic applications.

## 1. Introduction

Graphene and graphene-related materials, such as graphene oxide (GO) and reduced graphene oxide (rGO), have attracted huge attention in recent years due to their outstanding properties and wide range of potential applications. Their unique properties such as large surface area and excellent thermal and electrical conductivity have rendered them useful for many applications in energy storage [1], supercapacitors [2], electronics [3], water purification [4], catalysis [5], and membranes [6]. In these applications, rGO is preferred to pristine graphene due to the higher potential of production at the industrial scale since rGO synthesis does not require complex methodologies and equipment [7]. There are numerous rGO production methods, including chemical vapor deposition (CVD) [8] and graphite oxidation followed by various reduction routes [9]. The graphite oxidation route is the more promising, scalable, and commercially feasible method to achieving high quality and large-sized rGO flakes. This is attributed to the fact that the precursor GO disperses well in several solvents and can form stable hydrocolloids [10,11,12]. Hence, this facilitates the assembly of various three-dimensional (3D) macrostructures such as porous aerogels and films [7].

GO is formed by the chemical intercalation and oxidation of graphite through the use of several strong oxidizing agents [13,14] which are then subsequently delaminated to produce monolayer GO [15,16]. To regain graphene-like properties, numerous GO reduction methods have been implemented to restore the π-conjugated structure in GO, thereby transforming GO into rGO. Various GO reduction methods include thermal annealing [17], radiation induced [18], solar [19], seriography [20], and chemical reduction via use of various reducing agents [21,22,23,24]. Out of all these methods, chemical reduction of GO stands out as it can be conducted at temperatures lower than 100 °C without the need for sophisticated experimental setups and restores the π-conjugated structure resembling graphene [25,26]. This makes production of rGO material highly scalable with graphene-like properties. Several chemical reducing agents used include hydrazine hydrate [22], metal hydrides [23], ascorbic acid [27], and hydrogen iodide (HI) [21,24]. This high scalability process could potentially enable the use of large area rGO macrostructures for practical applications in electronics [28].

Moreover, the lateral size of GO plays a crucial role in its inherent properties and physical performance of the resultant rGO materials. Compared to small-sized GO, larger GO sheets possess low density of intersheet junctions, strong alignment propensity, and enhanced intersheet interactions [29], which facilitate the formations of subsequent macrostructures and nanocomposites to exhibit superior performance [30,31,32]. For example, it was shown that large-sized GO produced rGO with better electrical and thermal conductivities and higher efficiency in load transfer [29,30]. Additionally, filtration membranes fabricated from lager-sized GO are more advantageous than small-sized GO by possessing lesser nanochannels due to reduced boundaries formed by low density inter-sheet junctions and strong mechanical properties [33]. Hence, it is hypothesized that chemical reduction of larger-sized GO may lead to rGO with better physical properties than small GO [34,35]. However, it is unclear whether there is a direct link between GO size and its chemical reduction efficiency or the physical properties of rGO, as methods to synthesize GO of controllable lateral sizes, especially large ones, are lacking.

Recently, we developed a 1-pyrenebutyric acid (1-PBA) assisted method to synthesize monolayer GO at various controllable lateral dimensions, ranging from 0.7 to 116 µm [36]. Thus in this study, we investigated if the initial lateral size of GO affects chemical reduction efficiency of GO in suspension and in film and the electrical conductivity of rGO film. Our X-ray diffraction (XRD) and electron microscopy (SEM) analyses show that the extent of chemical reduction of GO in solution (suspension) increases with the lateral size of the GO sheets. In contrary, the reduction efficiency of GO films decreases with the size of GO sheets due to restacking and agglomeration of the GO in films and thus limitation of reductant diffusion. To overcome the problem of GO restacking in the GO film, we premixed large-sized GO (LGO) and small-sized GO (SGO) at different ratios for film preparation. The results reveal that the reduction efficiency of GO film is affected not only by GO size but also the interlayer spacing and restacking of GO. An optimal reduction was achieved with the film made of LGO and SGO at 3:1 (*w*/*w*), which exhibited the highest electrical conductivity of 85,283 S/m. Furthermore, our results show that the electrical conductivity of rGO films is linearly dependent on the C/O and I_D_/I_G_ ratio of the rGO films rather than the initial size of GO.

## 2. Results and Discussion

### 2.1. Characterisations of Three Different-Sized GO Samples

Three different sized GO samples were synthesized using our recently developed 1-PBA-assisted method [36]. Three different concentrations of KMnO_4_ (3×, 6×, and 10× graphite weight) were used to control the lateral size of the GO, as described previously [36]. These GO samples were named large GO (LGO), medium GO (MGO), and small GO (SGO), respectively, according to their relative sizes. Using AFM and FESEM analyses, we determined the average lateral dimension of LGO, MGO, and SGO to be 116.09 µm ± 42.65, 49.09 µm ± 23.08, and 0.72 µm ± 0.36, respectively (Table 1, Appendix A, online) [36]. The populations of single-layer GO in the SGO, MGO, and LGO samples were 91%, 98%, and 99%, respectively (Table 1, Appendix A, online), showing that SGO has the least percentage of monolayer GO. Thus, SGO flakes have the highest mean thickness of 1.88 nm as compared to MGO (1.34 nm) and LGO (1.24 nm). Consistently, more folds and wrinkles were observed for both LGO and MGO, while SGO flakes appeared to be flat due to their small lateral size (Appendix A, online).

High-resolution XPS spectra were used to assess the content of various oxygenated functional groups of the GO samples. XPS analysis showed that LGO has the highest mean C/O ratio of 3.58, while MGO and SGO have lower C/O ratios of 2.59 and 2.05, respectively (Table 1). This is due to carbonyl and carboxyl groups preferentially attacking the edges of graphite during oxidation, thus SGO has more edges [37]. Furthermore, Raman spectroscopy, a non-destructive and commonly used technique for structurally characterizing carbon materials [38], was conducted on the middle area of GO samples. The Raman spectrum of graphene-related materials generally shows two major features, the G band (E_2g_ symmetry of sp^2^ carbon atoms) at approximately 1575 cm^−1^ and the D band (breathing mode of A_1g_ symmetry) at around 1350 cm^−1^ [39]. As shown in Table 1 and Appendix A, LGO has the lowest I_D_/I_G_ ratio (intensity ratio of the D peak to the G peak) of 0.81, compared with MGO (0.88) and SGO (0.93), and thus the lowest structural defect [36]. This is inconsistent with previous studies which showed that the degree of graphite oxidation increases with oxidant concentration [36,40,41,42]. KMnO_4_ oxidant causes sp^3^-hybridized epoxy C–O groups to distort the basal plane of graphene, inducing a greater degree of stretching and breaking of the underlying C–C bonds and, subsequently, fragments of GO sheets [36,40,41,42]. In summary, LGO sheets have the highest lateral dimension and C/O ratio along with the lowest Raman I_D_/I_G_ ratio and structural defect compared to MGO and SGO (Table 1, Appendix A, online).

### 2.2. Effects of GO Size on Its Reduction in Suspension

To understand if the initial size of GO affects the reduction efficiency of GO, we tested GO reduction using HI acid as reductant in suspension. Upon addition of the reductant, a color change from brown to black was observed, showing the transformation of GO suspension to rGO colloids. The GO and corresponding rGO samples (r-LGO, r-MGO, r-SGO) were characterized using high-resolution XPS to assess the content of various oxygenated functional groups and C/O ratios of the rGO samples (Figure 1, Table 2). Firstly, C/O ratios of all three rGO samples increased upon chemical reduction compared with that of GO from in a range of 2–3.6 to 15–20 (Table 1 and Table 2) as expected. C1s deconvolution analyses suggested that the majority of oxygen-containing groups (e.g., hydroxyl and epoxy groups) in GO samples (Figure 1b–d) were almost completely removed, and both the C=C and C–C bonds became dominant in the rGO colloids (Figure 1e–g). This was evidenced by the presence of one single peak with a narrow tail in the higher binding energy region of C1s spectra of rGO (Figure 1e–g). As shown in Table 2, HI acid reduced content of C–O bonds from about 30–40% in GO to 10–20% in rGO colloids, enabling greater restoration of sp^2^ carbon as shown by the increase in C=C bonds. Additionally, partial reduction of carbonyl C=O groups were observed as all rGO samples had a lower percentage of C=O groups as compared to the corresponding GO samples (Table 2, Appendix A, online). Furthermore, residual iodine, due to the presence of I3d peaks detected at 619eV (I3d 5/2) and 630.8eV (I3d 3/2) [43], was observed in all rGO samples after washing with water (Figure 1a). This is in agreement with previous studies [21,44], showing the rGO reduced by HI are doped with iodine, possibly due to the iodination of alcohol groups. Secondly, among these three rGO samples, r-LGO has the highest C/O ratio of 20.12, followed by r-MGO (18.93), and then r-SGO (15.24). Compared with the corresponding GO samples, the C/O ratio increased by 16.5, 16.3, and 13, respectively, in r-LGO, r-MGO, and r-SGO. Thus, XPS analysis showed that reduction of LGO is the most effective, followed by MGO and then SGO.

Next, Raman spectroscopy was used to characterize these rGO samples. Compared to GO, the I_D_/I_G_ ratio of rGO increased due to the increase in sp^2^ carbon because of chemical reduction, which alters the structure of precursor GO [38,45]. As shown in Figure 2b, all rGO colloidal samples have a higher I_D_/I_G_ ratio than GO as expected. Moreover, r-LGO had the highest I_D_/I_G_ ratio of 1.33 as compared to r-MGO (1.29) and r-SGO (1.21), and the increase in I_D_/I_G_ ratio (I_D_/I_G_) was 0.52, 0.41, and 0.28, for r-LGO, r-MGO, and r-SGO, respectively (Table 2). The trends of ID/IG ratio before and after GO suspension reduction are different. This may be due to the fact that the LGO suspension undergoes the largest extent of chemical reduction (followed by MGO), which leads to a higher proportion of smaller-sized sp2 domains and increases the proportion of graphitic edges [46,47]. Furthermore, the G-band of the rGO samples (1565–1568 cm^−1^) shifted towards lower wavelength in comparison to that of GOs (1570–1577 cm^−1^) with r-LGO shifted most due to removal of oxygen content and restoration of delocalized π conjugation graphitic structure [48]. Similarly, the D-band rGO also blue-shifted but only slightly. Thus, the Raman analysis is consistent with that of XPS, also suggesting that LGO has the highest chemical reduction efficiency followed by MGO and then SGO under the experimental conditions. This size-dependent result is consistent with previous studies [49,50,51]. The presence of a shoulder peak near and at right side of the G-band in the GO and rGO samples is probably due to the presence of defected graphitic structures formed during chemical oxidation and reduction processes, respectively [52,53]. In addition, we also obtained such a size-dependence of GO reduction using L-ascorbic acid, an eco-friendlier compound, as reductant (Appendix A, online). In summary, larger-sized GOs deoxygenated more efficiently than smaller-sized GOs upon chemical reduction in suspension. This is probably due to two reasons: (1) LGO has a relative low density of structural defects, and thus it is relatively easier for restoration of delocalized π conjugation graphitic structure and (2) has the presence of the highest (99%) proportion of monolayer GO in LGO enabling the highest accessibility to reductants.

### 2.3. Effects of GO Size on Its Reduction in Film

It was shown that efficient chemical reduction of GO is limited not only by the property of chemical reductants [44] but also by restacking and aggregation of GO sheets via inter-sheet π–π interactions [54,55,56]. However, it is unknown whether and how GO size affect its chemical reduction in forms of films. This is important since highly reduced rGO films have the potential to be used within electronic applications due to being freestanding, lightweight, and highly electrically conductive [7]. Thus, we prepared GO films using the above three different-sized GOs and chemically reduced them by immersion of the GO films in 55% HI acid solution. XPS and Raman analyses were used to characterize the resulting rGO films, and the results are shown in Figure 3 and Table 2. Interestingly, both XPS and Raman analyses suggested an opposite GO size-dependence compared with GO reduction in suspension. XPS (Figure 3a, Table 2) showed that r-SGO film was more deoxygenated with a higher mean C/O ratio (16.2) than that of r-LGO (12.72) and r-MGO films (14.63). XPS C1s spectra deconvolution suggested that r-SGO films (Figure 3d) have relatively higher percentage of hybridized sp^2^ and sp^3^ bonds along with fewer epoxides and carboxylic functional groups than r-LGO (Figure 3b) and r-MGO films (Figure 3c). Consistently, Raman analyses showed that I_D_/I_G_ ratio of r-SGO film was 1.27, higher than r-LGO (1.14) and r-MGO (1.20) films (Table 2), suggesting that GO reduction in film is least efficient for LGO, followed by MGO and SGO (Figure 3f). Thus, GO reduction efficiency decreased when the lateral size of the GO increased. A similar size-dependent result was also obtained using L-ascorbic acid as reductant (Appendix A, online).

Taken together, our results suggest that there is a different GO size-dependence between chemical reduction of GO in suspension and in film. In contrary to the trend of GO reduction in suspension, the reduction effectiveness of GO films decreases as the lateral size of GO increases. We reasoned this difference may result from the agglomeration of LGO in film since GO films are formed by restacking of monolayer GO sheets. The LGO sheets may tend to restack and form more compacted films due to higher sp^2^ hybridized conjugation regions, although they are well dispersed in suspension. Thus, we hypothesized that a possible way to increase chemical reduction efficiency of larger GO film is to decrease stacking of LGO sheets in film.

### 2.4. Positive Effects of SGO on Reduction Efficiency of LGO Film and the Mechanism

Increasing effectiveness of chemical reduction in GO films is of paramount importance as highly reduced free-standing rGO films have high electron mobility capabilities which are very desirable for various electronic applications. Reduction of macrostructural 3D GO film has been shown to be diffusion-controlled [21], in which reductants reduce GO film from the outer to inner part. Hence, we reasoned that when cross-section of GO films widened to allow effective diffusion of the chemical reductant, the reduction efficiency of GO film would increase. Furthermore, the relative large surface area and high degree of oxidation of SGO decrease the tendency of SGO to restack (compared with LGO) during GO film. Thus, we hypothesized that addition of SGO to LGO or MGO films may increase the chemical reduction efficiencies of the subsequent L/SGO or M/SGO films. To test this hypothesis, SGO was mixed with LGO (or MGO) in a ratio of 1:3, 1:1, and 3:1 by weight (*w*/*w*), respectively, and then vacuum-filtrated. The formed L/SGO films were reduced using HI acid as above to form corresponding r-L/SGO films.

XPS analyses (Figure 4a) revealed that the C/O ratio of all three mixed r-L/SGO films was higher than r-SGO or r-LGO alone, and it increased with increase in LGO content (Figure 4c). The r-L/SGO (3:1) film showed the highest C/O ratio of 20.4. Raman spectroscopy analysis of the r-L/SGO films (Figure 4b) showed that the change of I_D_/I_G_ ratio correlated well with C/O ratio change (Figure 4c). It also increased with the increase in LGO percentage, and r-L/SGO (3:1 *w*/*w*) displayed the highest I_D_/I_G_ (1.34), higher than both SGO and LGO alone. A good linear correlation between C/O and I_D_/I_G_ ratios was observed (Figure 4d). Moreover, similar results were obtained for mixing MGO and SGO together as well as for using L-ascorbic acid as reductant (Appendix A, online).

To understand why the L/SGO film (made of LGO and SGO) was better reduced than the films made of LGO or SGO alone, we performed structural analyses of these films using FESEM, XRD, and Raman spectroscopic methods. First, FESEM images of the GO films (Figure 5a–c showed that the cross-section thickness of LGO film (18.6 µm) was the lowest in comparison with L/SGO (3:1) (20.5 µm) and SGO (24.9 µm). This is consistent with our hypothesis that the LGO sheets may tend to restack due to higher sp^2^ hybridized conjugation regions and thus form more compact film. Upon chemical reduction, the cross-section thickness of corresponding rGO films (Figure 5e–h) decreased to 5.09 µm for r-LGO, 3.46 µm for r-L/SGO (3:1 *w*/*w*), and 4.39 µm for r-SGO, showing that r-LGO film had the thickest cross-section among the three rGO films, while r-L/SGO (3:1 *w*/*w*) film had the lowest thickness.

Next, XRD analysis of GO films (Figure 5i) showed that the peaks centered at 9.85° (SGO), 10.65° (L/SGO), and 11.28° (LGO), corresponding to an interlayer d-spacing of 0.897, 0.83, and 0.783 nm, respectively, with the diffraction peak of LGO most right-shifted. It confirmed FESEM result that LGO film had a more compact structure compared to SGO and L/SGO. Upon chemical reduction, all XRD spectra of rGO films (Figure 5j) displayed only one diffraction peak at 2θ around 24–26°. The peaks were centered at 25.1° for r-L/SGO (3:1 *w*/*w*), 24.4° for r-SGO, and 24° for r-LGO, respectively, with corresponding d-spacing to be 0.354, 0.364, and 0.37 nm. The r-L/SGO (3:1) film was the most right-shifted and thus had the most compact structure among these rGO films. A previous report of GO film reduction by HI acid showed highly reduced GO with 2θ at 24.6° and interlayer distance ~0.362 nm [57]. Our r-L/SGO (3:1 *w*/*w*) shows an even smaller interlayer distance, indicating a more efficient chemical reduction. In addition, it is known that a greater intensity of Raman 2D peak represents higher density of sp^2^ carbon in rGO films [58]. As shown in Figure 5k, the 2D Raman peak of r-L/SGO (3:1 *w*/*w*) displayed the highest relative intensity compared to both r-SGO and r-LGO, suggesting greater graphitization and restoration of sp^2^ carbon in r-L/SGO (3:1 *w*/*w*) film. Taken together, our analyses showed that addition of SGO increases the cross-section thickness and interlayer d-spacing of LGO film and thus its chemical reduction efficiency. Consequently, the mixed r-L/SGO (3:1) film displayed the lowest thickness, most compact structure, and greatest restoration of sp^2^ carbon because of the increased reduction efficiency.

### 2.5. Electrical Conductivity of rGO Films

Next, the electrical conductivity of various rGO films was measured using a non-contact sheet resistance meter three times and the electrical conductivity of individual films was calculated (see method). As shown in Figure 6a, the electrical conductivity increased with the increase in LGO in the mixture, and r-L/SGO (3:1) films exhibited the highest electrical conductivity of 85,283 S/m out of all rGO films. This electrical conductivity value is the highest reported (Appendix A, online), to the best of our knowledge, and is superior to rGO films synthesized by thermal annealing [59] and electrochemically produced graphene [60,61]. Furthermore, a good linear correlation between electrical conductivity and both C/O and I_D_/I_G_ ratio was obtained (Figure 6b–c). Moreover, a similar trend of LGO content dependence was obtained using L-ascorbic acid as reductant (Appendix A, online).

Taken together, our results showed that the electrical conductivity of rGO film was dependent on the C/O ratio of rGO, rather than the lateral size of GO used, and r-LGO film displayed lower electrical conductivity than r-SGO film. This is contrary to the prediction that large rGO may have higher electrical conductivity due to the lower densities of inter-sheet junctions [35]. We showed that although LGO was easier to be reduced than SGO in suspension, the LGO film was harder to be chemically reduced than SGO film. This is because LGO is more prone to self-interaction than SGO, so the interlayer spacing of LGO sheets is small and the structure of LGO film is relatively more compact than that of SGO film. The compact structure of LGO film can limit the diffusion of reductant into the film and thus decrease its reduction efficiency. Furthermore, we showed that the film made using mixed-sized GO samples provide a way to increase reductant accessibility and thus LGO reduction efficiency in film, consequently increasing the electrical conductivity of the rGO film. We reason this is because SGO is able to minimize the agglomeration of LGO and enhance interlayer distance of GO films and consequently increase its reduction efficiency and r-L/SGO electrical conductivity. In addition, the lesser inter-sheet junction within LGO could have also contributed to increase electrical conductivity. In this study, r-L/SGO (3:1) film, a film made of LGO and SGO at 3:1 weight ratio exhibited the highest electrical conductivity of 85,283 S/m. Moreover, the positive linear correlation between electrical conductivity and I_D_/I_G_ ratio can be reasoned by the Tuinstra–Koenig relation as a measure for the size of sp^2^ domains [38] in which chemical reduction of GO results in the formation of small patches of graphene within holes of the initial GO structure [62,63]. As a result, higher I_D_/I_G_ ratio emerges as a direct consequence of the increased number of smaller graphene-like regions. This induces higher electron mobility and thus greater electrical conductivity within the resulting graphitic-like lattice structure. Finally, similar to the reduction efficiency, our results show that the electrical conductivity of the rGO-based films can be tuned by mixing GO flakes with different lateral dimensions, which contributes to the electrical conductivity differently.

## 3. Experimental Work

### 3.1. Materials

Flake graphite (>80% 100-mesh), potassium permanganate (KMnO_4_) (>99.5%), and L-ascorbic acid were provided by Sigma-Aldrich. Concentrated sulfuric acid (H_2_SO_4_, 95%) was obtained from Fisher Scientific, Loughborough, UK. The 55% HI acid was purchased from Aladdin-reagent Inc. and used as received. 1-pyrenebutyric acid (1-PBA, 97%) and 30% hydrogen peroxide (H_2_O_2_) were from Alfa Aesar. All chemicals were of analytical grade and used as received.

### 3.2. GO Synthesis

Three different lateral-size GOs (LGO, MGO SGO) were prepared using 1-pyrenebutyric acid-assisted method according to our previous report [36]. Typically, 100 mg 1-pyrenebutyric acid (1-PBA) was added to 50 mL of concentrated sulfuric acid (95%) and mechanically stirred prior to addition of 1 g of graphite flakes. The subsequent 1-PBA-graphite mixture was then left to stand for 2 days at room temperature as this was the optimal duration for complete graphitic oxidation to form GO as determined previously [36]. Simultaneously, the acidified oxidants were synthesized in a separate mixture of 3 g of KMnO4 added to 50 mL of 95% sulfuric acid under mechanical stirring. The oxidizing agents were then added slowly to the 1-PBA-graphite mixture under mechanical and left to stand overnight at room temperature. Lastly, cold-MQ water was added slowly to the mixture and the unreacted purple paste of KMnO_4_ were removed with H_2_O_2_. The supernatant was carefully drafted off, then MQ water was added to wash the GOs. This process was continued until pH of the GO sample increased to about 6. This GO is named large GO (LGO) for its large lateral flake size [36]. In separate experiments, the corresponding medium-sized (MGO) and small-sized (SGO) GO were synthesized in the presence of 6 g and 10 g KMnO_4_ (instead of 3 g).

### 3.3. Physical Characterisations

Cross-sections of rGO films were imaged by field emission scanning electron microscopy (FESEM), using FESEM FEI Quanta 250 at an accelerating voltage of 10 kV. The GO and rGO films were mounted using carbon tabs and clipped in an upright position to image their cross-section for thickness measurement.

Renishaw Invia with a 525 nm laser and 50× objective was used to record Raman spectra. The vacuum-filtrated rGO films were attached on a microscope glass slide by adhesive tape. The Raman spectra were fitted using linear background subtraction and Voigt functions.

PANalytical X’Pert Pro X’Celerator diffractometer with a Cu Kα radiation (λ = 1.5406 Å) at step size of 0.02° and 5 s per step were used for X-ray diffraction (XRD) measurements. Interlayer spacing of GO and rGO was calculated using Bragg’s law: λ = 2*d* sin(θ), where λ is X-ray wavelength, *d* is the interlayer spacing of GO or rGO, and θ is the diffraction angle.

X-ray photoelectron spectroscopy (XPS) was carried out using an Axis Ultra Hybrid spectrometer (Kratos Analytical, Manchester, United Kingdom), using monochromatic Al Kα X-ray radiation at 1486 eV (10 mA emission at 15 kV, 150 W) under ultra-high vacuum at a base pressure of 1 × 10^−8^ mbar. GO and rGO samples were pressed onto conductive copper tape. High resolution C1s and O1s spectral deconvolution was performed using CASAXPS software (Casa Software Ltd., Devon, UK) with Shirley-type backgrounds subtraction. Gaussian–Lorentzian functions were fitted to all identified functional groups, which are constrained to the following binding energies: C–C and C=C at 284.5–284.6 eV; C–O at 285.5–286.6 eV (C1s) and 532–533 eV (O1s); O–C=O at 288.6–290 eV (C1s) and 533–534 eV (O1s); and π–π* at 290–292 eV.

Electrical conductivities of rGO films was measured by using a benchtop non-contact sheet resistance meter (Jandel 20J3 Sensor) to obtain sheet resistance of rGO films via eddy currents under ambient conditions. The electrical conductivities of rGO films were calculated based on:(1)$$\sigma =\frac{1}{R\times T}$$
where σ, *R*, and *T* represent electrical conductivity (S/m), measured sheet resistance (Ω/sq) and thickness (m), respectively. For each film, three values were obtained at different points to get an average sheet resistance value.

### 3.4. Preparation of GO Films

Briefly, 10 mL GO solution (2 mg/mL) was prepared for vacuum filtration on polytetrafluoroethylene (PTFE) membrane (Millipore 0.2 μm pore size), used as received, to synthesize various PTFE-GO films. Then, 70% ethanol was used to remove dried GO from the PTFE membrane, which was further dried in vacuum oven at room temperature overnight.

### 3.5. Preparation of rGO Films

Free-standing GO films were subsequently submersed in L-ascorbic acid (15× *w*/*t*) at 90 °C for 1 h to form rGO films. For HI reductant, the GO films were immersed in 55% HI within a sealed cuvette at 85 °C for 1 h. Eventually, the rGO films were washed with MQ water several times until of neutral pH with clear residual solution and dried in vacuum at room temperature overnight.

## 4. Conclusions

In this study, we investigated how GO lateral size and its oxidation degree affects GO chemical reduction efficiency in suspension and in film, respectively, as well as the electrical conductivity of rGO films. Our results showed that there is a different GO size and oxidation dependence between chemical reduction of GO in suspension and film. In suspension, GO reduction efficiency increases with the lateral size of GO of lower oxidation extent. However, the reduction efficiency of GO films decreases as the size of the GO sheet increases due to the restacking and agglomeration of large-sized GO in film and the presence of lesser oxygenated functional groups. Next, we showed that addition of SGO to LG prevented LGO from restacking or agglomeration and consequently increased the chemical reduction efficiency of the GO film. An optimal reduction was achieved in the film made of LGO and SGO at 3:1 (*w*/*w*) ratio. Furthermore, we report that r-L/SGO (3:1) films exhibited the highest electrical conductivity of 85,283 S/m, out of all rGO films reported so far. Finally, our results showed that the electrical conductivity of rGO film is linearly dependent on the C/O ratio and I_D_/I_G_ ratio of rGO films, rather than the initial size of GO. We believe that the strategy described in this study is valuable in potentially achieving large-scale production of electrically conductive graphene films via the chemical reduction of GO films.

## Figures and Tables

**Figure 1 molecules-27-07840-f001:**
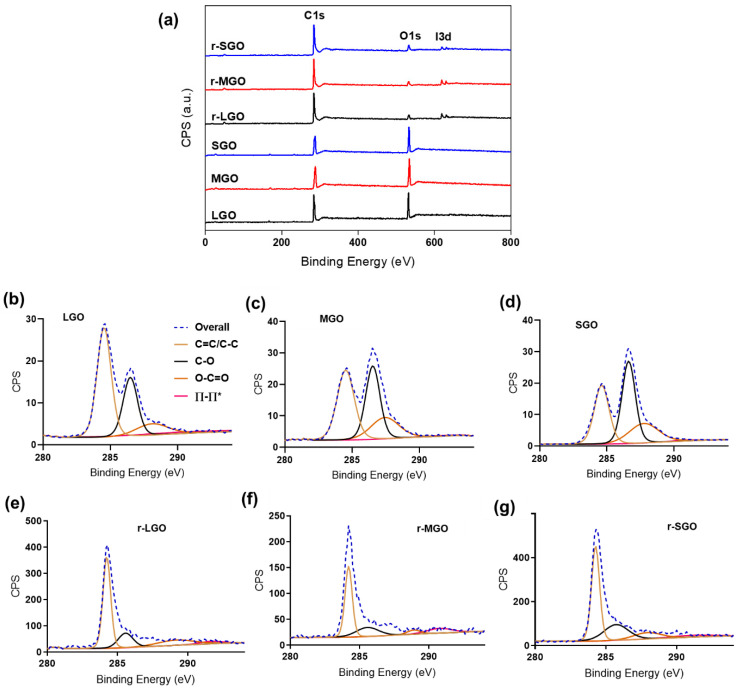
XPS characterization of rGO prepared using different-sized GO in suspension. (**a**) General survey XPS spectra of three GO samples and corresponding rGO prepared using HI acid as reductant. (**b**–**g**) XPS C1s deconvolution spectra of (**b**) LGO, (**c**) MGO, (**d**) SGO, (**e**) r-LGO, (**f**) r-MGO, and (**g**) r-SGO.

**Figure 2 molecules-27-07840-f002:**
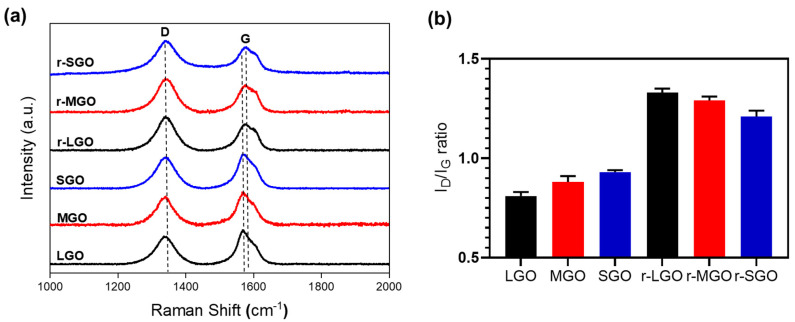
Raman spectroscopy of three GO and rGO colloidal samples. (**a**) Normalized general Raman spectra of different-sized GO and corresponding rGO. (**b**) I_D_/I_G_ ratios based on the Raman intensity of D band and G band. Error bars represent standard error of mean (SEM), n = 3.

**Figure 3 molecules-27-07840-f003:**
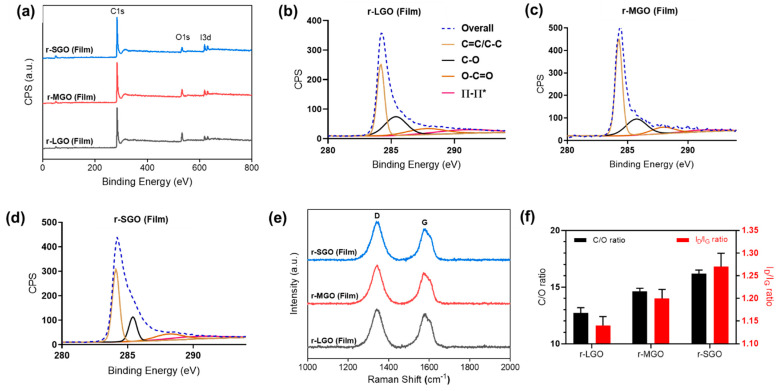
Chemical composition and physical characterization of rGO films. (**a**) General survey XPS spectra of rGO films. r-SGO (blue), r-MGO (red), and r-LGO (black). (**b**–**d**) XPS C1s deconvolution spectra of r-LGO, r-MGO, and r-SGO, respectively. (**e**) Raman spectra of rGO films. (**f**) Correlation between C/O, I_D_/I_G_ ratio, and various rGO films. All GO films were reduce using HI acid as reductant. Error bars represent standard error of mean (SEM), n = 3.

**Figure 4 molecules-27-07840-f004:**
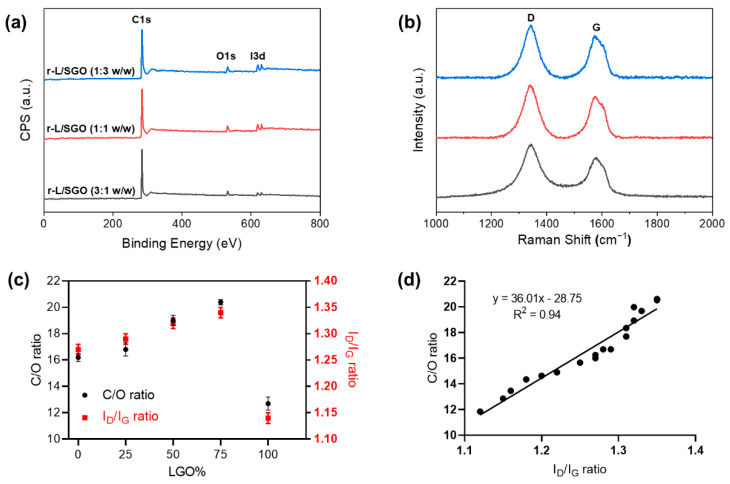
Characterizations of rGO films prepared with mixed LGO and SGO at different ratios. All rGO film prepared with HI acid as reductant. (**a**) XPS spectra; (**b**) Raman spectra; (**c**) dependence of C/O and I_D_/I_G_ ratios of rGO films on LGO percentage used in the film preparations. All error bars represent standard error of mean, n = 3. The solid line in (**c**) represents a linear regression fit. (**d**) Correlation between C/O ratio and Raman I_D_/I_G_ ratio.

**Figure 5 molecules-27-07840-f005:**
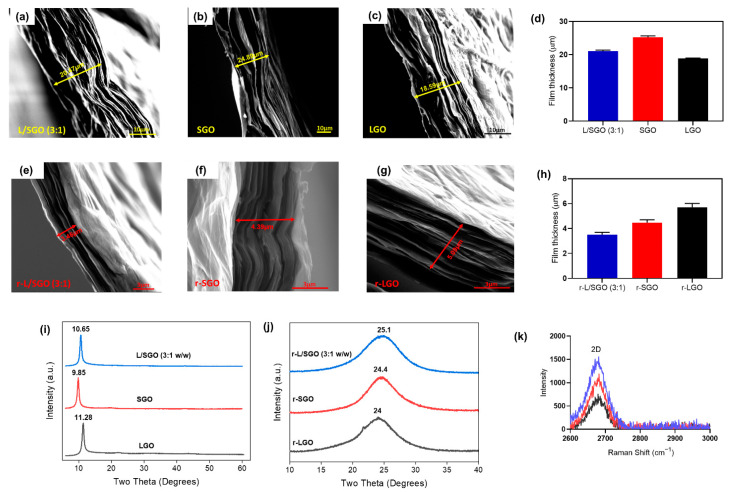
Structural characterization of different GO and rGO films. (**a**–**c**): FESEM images of the cross-section of L/SGO (3:1), SGO, and LGO films, respectively. (**d**) Bar chart plot of cross-sectional film thickness for various GO. (**e**–**g**): FESEM images of the cross-section of r-L/SGO (3:1 *w*/*w*), r-SGO, and r-LGO films, respectively. (**h**) Bar chart plot of cross-sectional film thickness for various rGO. (**i**) XRD spectra of various GO films. (**j**) XRD spectra of various rGO films. (**k**) Raman spectra 2D band of various rGOs. The error bars represent SEM with n = 3. All rGO films ware prepared use HI acid as reductant. A comparative study using L-ascorbic acid reductant was shown in Appendix A.

**Figure 6 molecules-27-07840-f006:**
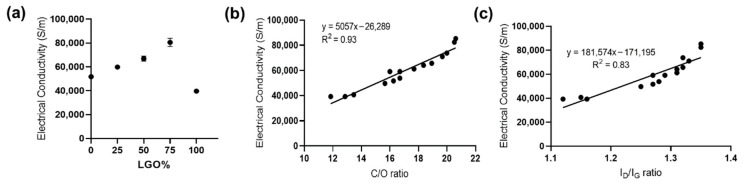
Electrical conductivity of rGO films. (**a**) LGO content dependence of the electrical conductivity of rGO films. All rGO films ware prepared use HI acid as reductant. The error bars represent SEM with n = 3. (**b**) Correlation between electrical conductivity and C/O ratio. The solid line represents a linear regression fit. (**c**) Correlation between electrical conductivity and Raman I_D_/I_G_ ratio. The solid line represents a linear regression fit.

**Table 1 molecules-27-07840-t001:** Properties of three different GO samples. Error bars represent standard error of mean (SEM), n = 3.

	Lateral Size	Monolayer% (≤2 nm)	Flake Thickness	C/O Ratio	I_D_/I_G_ Ratio
LGO	116.09 µm ± 42.65	99	1.24 ± 0.1 nm	3.58 ± 0.23	0.81 ± 0.02
MGO	49.09 µm ± 23.08	98	1.34 ± 0.05 nm	2.59 ± 0.10	0.88 ± 0.03
SGO	0.72 µm ± 0.36	91	1.88 ± 0.04 nm	2.05 ± 0.30	0.93 ± 0.01

**Table 2 molecules-27-07840-t002:** Summary of XPS and Raman spectra analysis of rGO samples prepared with different size GO in suspension and film. Error bars represent standard error of mean (SEM), n = 3.

		C/O Ratio	C=C/C–C (%)	C–O (%)	O–C=O (%)	π–π (%)	I_D_/I_G_ Ratio
Suspension	r-LGO	20.1 ± 0.1	60.49	11.48	13.44	14.58	1.33 ± 0.02
	r-MGO	18.3 ± 0.3	57.65	16.12	14.18	12.04	1.29 ± 0.02
	r-SGO	15.6 ± 0.2	54.59	23.45	16.57	5.38	1.21 ± 0.03
Film	r-LGO	12.7 ± 0.5	53.49	27.26	13.16	6.09	1.14 ± 0.02
	r-MGO	14.6 ± 0.3	52.66	27.6	14.14	5.6	1.20 ± 0.02
	r-SGO	16.2 ± 0.3	55.03	25.99	13.27	5.71	1.27 ± 0.03

## Data Availability

The datasets generated during and/or analyzed during the current study are available from the corresponding author on reasonable request.

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
