# Peer review of "The Influence of Lateral Size and Oxidation of Graphene Oxide on Its Chemical Reduction and Electrical Conductivity of Reduced Graphene Oxide"

_molecules, 2022, doi:10.3390/molecules27227840_

Round 1
Reviewer 1 Report
1. In the keywords of the article, recommend to remove “Atomic force microscopy; Field emission scanning electron microscopy; X-ray photoelectron 25spectroscopy; X-ray diffusion”, these testing methods are general methods for characterizing material properties, which are not unique to this paper.
2. In this paper, more KMnO4 was added to oxidize flake graphite to prepare GO, this method will lead to the destruction of the surface layer of the regular grahite, thereby reducing its conductivity of rGO. Because reduction cannot re-establish the original graphite structure on the surface of the rGO.
Therefore, it is suggested that the title of the article is the influence of the degree of oxidation on the conductivity of reduced grapheme, rather than the size of graphene oxide.
Author Response
We thank the reviewer for your comments that helped us to improve the quality of the manuscript. Please see our pint-by-point response below.
- In the keywords of the article, recommend to remove “Atomic force microscopy; Field emission scanning electron microscopy; X-ray photoelectron 25spectroscopy; X-ray diffusion”, these testing methods are general methods for characterizing material properties, which are not unique to this paper.
Answer 1: We have removed these keywords and replaced them with ‘graphene-related materials and films’.
- In this paper, more KMnO4 was added to oxidize flake graphite to prepare GO, this method will lead to the destruction of the surface layer of the regular grahite, thereby reducing its conductivity of rGO. Because reduction cannot re-establish the original graphite structure on the surface of the rGO. Therefore, it is suggested that the title of the article is the influence of the degree of oxidation on the conductivity of reduced grapheme, rather than the size of graphene oxide.
Answer 2: Thanks for the reviewer’s suggestion, we have changed the title to ‘The influence of lateral size and oxidation of graphene oxide on its chemical reduction and electrical conductivity of reduced graphene oxide’. We also modified the abstract and conclusion accordingly.
Reviewer 2 Report
In their study, the authors investigated how GO lateral size affects GO chemical reduction efficiency in suspension and in film, respectively, as well as the electrical conductivity of rGO films. They showed that in suspension, GO reduction efficiency increases with the lateral size of GO. However, the reduction efficiency of GO films decreases as the size of GO sheet increases due to restacking and agglomeration of LGO in film. They concluded that an optimal reduction achieved in the film comprised by LGO and SGO at 3:1 (w/w) ratio and the film of this composition exhibited the highest electrical conductivity of 85,283 S/m. Besides, they showed that the electrical conductivity of rGO film is linearly dependent on the C/O ratio and ID/IG ratio of rGO films, rather than the initial size of GO.
Critical comments:
Line 40. GO is not a dispersant. It is a dispersed phase in the solvent, being a dispersion medium.
Line 46. ‘reduction’ should be changed for ‘reduction methods’.
Line 146. The total number of the GO sheets measured for LGO and MGO particles (N = 20) is not enough for statistical analysis.
Figs. 2a, 3e, and 4b. In these figures, the G band is superimposed with a blue-shifted band, which origin is unknown and therefore should be clarified and discussed.
Lines 146-153. The 2.6 section should rather be moved to the 2.3 section.
Author Response
We thank the reviewer for your comments that helped us to improve the quality of the manuscript. Please see our point-by-point response below.
- Line 40. GO is not a dispersant. It is a dispersed phase in the solvent, being a dispersion medium.
Answer: Thanks to the reviewer, we have amended the sentence as ‘This is attributed to the fact that precursor GO disperses well in several solvents and can form stable hydrocolloids [10-12].’
- Line 46. ‘reduction’ should be changed for ‘reduction methods’.
Answer: We amended it to ‘reduction methods’.
- Line 146. The total number of the GO sheets measured for LGO and MGO particles (N = 20) is not enough for statistical analysis.
Answer: We think that the reviewer may mix it up with our previous publication, as in line 146 and the whole manuscript, there is not a statistical analysis with N=20.
- Figs. 2a, 3e, and 4b. In these figures, the G band is superimposed with a blue-shifted band, which origin is unknown and therefore should be clarified and discussed.
Answer: This was discussed in lines 247-250 ‘The presence of a shoulder peak near and at right side of the G-band in the GO and rGO samples is probably due to the presence of defected graphitic structures (e.g. lattice vacancies, carbon bond strains) formed during chemical oxidation and reduction processes respectively [52, 53.]’
Lines 146-153. The 2.6 section should rather be moved to the 2.3 section
Answer: Thanks for the suggestion; we have changed this as suggested.
Reviewer 3 Report
Please see the attached file.

Author Response
Please find the reply is attached.

Round 2
Reviewer 3 Report
The MS can be published now.